# Microstructure and Mechanical Properties of High Relative Density γ-TiAl Alloy Using Irregular Pre-Alloyed Powder

Mengjie Yan [1], Fang Yang [1,2,*], Boxin Lu [1], Cunguang Chen [1,2], Yanli Sui [2,3] and Zhimeng Guo [1,2,*]

1   Institute for Advanced Materials and Technology, University of Science and Technology Beijing, Beijing 100083, China; b20180554@xs.ustb.edu.cn (M.Y.); luboxin2008@outlook.com (B.L.); cgchen@ustb.edu.cn (C.C.)
2   Innovation Group of Marine Engineering Materials and Corrosion Control, Southern Marine Science and Engineering Guangdong Laboratory (Zhuhai), Zhuhai 519082, China; yls@ustb.edu.cn
3   State Key Laboratory for Advanced Metals and Materials, University of Science and Technology Beijing, Beijing 100083, China
*   Correspondence: yangfang@ustb.edu.cn (F.Y.); zmguo@ustb.edu.cn (Z.G.)

**Abstract:** Preparing high relative density γ-TiAl alloy by pressure-less sintering at low-cost has always been a challenge. Therefore, a new kind of non-spherical pre-alloyed TiAl powder was prepared by the reaction of TiH$_2$ powder and Al powder at 800 °C to fabricate high-density Ti-48Al alloy via pressure-less sintering. The oxygen content was controlled to below 1800 ppm by using coarse Al powder (~120 μm). The sintered densities ranged from 92.1% to 97.5% with sintering temperature varying from 1300 °C to 1450 °C. The microstructure of the sintered compact was greatly influenced by the sintering temperature. The as-sintered samples had a near-γ structure at 1350 °C, a duplex structure at 1400 °C, and a nearly lamellar structure at 1450 °C. To achieve full densification, non-capsule hot isostatic pressing was performed on the 1350 °C and 1400 °C sintered samples. As a result, high compressive strengths of 2241 MPa and 1931MPa were obtained, which were higher than the existing Ti-48Al alloys.

**Keywords:** γ-TiAl alloy; pressureless sintering; relative density; microstructure

## 1. Introduction

γ-TiAl-based alloys with low density, high mechanical properties at elevated temperatures, excellent creep performance and good corrosion resistance have been widely used in the recent decades, which are considered one of the most potential materials for aerospace and automobile industries [1–4]. In the aerospace industry, γ-TiAl can be used instead of Inconel 718 to prepare components, such as turbine blades and compressor blades. In the automotive industry, these materials are suitable for racing and high-end vehicle parts, such as engine valves, turbine wheels and connecting rods [5,6]. For example, Ti-48Al-2Cr-2Nb alloy has been used to manufacture the six-stage and seven-stage low pressure turbine blades in GEnx engines [7]. However, TiAl alloys generally suffer from low ductility at room temperature, resulting in the manufacturing difficulties. Some reports have found that adjusting the grain size in a suitable range and improving the homogeneity can be beneficial for improving the workability of this alloy [8–10].

In recent years, powder metallurgy (PM) technique is of special interest because of no-composition segregation, uniform microstructure, fine grain and low cost [11–13]. The economical cold-compact and pressure-less sintering process is one of the most attractive PM fabrication approaches for γ-TiAl [14], in which the raw powder is the most important part of pressure-less sintering.

Two types of TiAl powders are often used: Pre-alloyed (PA) powder, prepared by gas atomization and rotating electrode atomization, and the blended elemental (BE) powder composed of Ti, Al and other alloying elements. PA powder with a mean particle size

of 40–100 μm is approximately spherical, and thus, results in much poor formability and sinterability [11,14–16]. In general, a binder is added into the PA powder to achieve formation and consolidation by hot isostatic pressing (HIP) [11,17], or spark plasma sintering (SPS) [18]. BE powder is more convenient by adding other alloy elements [19,20]. However, Ti and Al react at a temperature above the melting point of Al, quickly releasing a lot of heat. This causes the volume expansion up to 30% [21], and leads to the occurrence of Kirkendall effect [11,22]. Therefore, pressure-assisted sintering is usually conducted to achieve higher sintering density [23–25]. It has been reported that the density of Ti-48Al alloy, prepared by SPS using BE powder, increases from 88% to 98.9%, compared to pressure-less sintering at 1350 °C [25]. To this end, it is difficult to use the above two types of TiAl powders to prepare high-density $\gamma$-TiAl by pressure-less sintering.

Furthermore, the high oxygen content is also a serious problem. As the oxygen content increases, the mechanical properties significantly decrease [26]. The high reactivity of TiAl, with oxygen, leading to oxygen pollution during the preparation process, cannot be completely avoided [27]. The solubility of oxygen in $\gamma$-TiAl is very low, less than 3 at.% [28]. In such cases, it is easy to form precipitates, such as rutile and alumina [29]. Therefore, in our previous works [30,31], powder transfer and powder handing are taken under Ar atmosphere in the entire procedure to control the interstitial content, especially oxygen content. Correspondingly, the oxygen content of Ti-23Al-17Nb with comprehensive mechanical properties is below 1500 ppm [15].

Therefore, there is interest in preparing high-density Ti-48Al alloy with low oxygen content by pressure-less sintering. In this study, a new kind of PA powder, with irregular shapes, was employed to prepare Ti-48Al. The irregularly shaped PA powder has better formability and sinterability, and was prepared by pre-sintering the $TiH_2$ and Al powders followed by grinding with strict process control for oxygen isolation. As a result, fully dense of Ti-48Al alloy was produced by pressure-less sintering. Besides, Al powders with different particle sizes were used to investigate the oxygen content change.

## 2. Experimental Procedures

### 2.1. Sample Preparation

Raw materials were $TiH_2$ powder produced by hydrogenation dehydrogenation process and Al powder produced by gas atomization. The particle sizes of the raw powders are shown in Table 1. Al powders with different particle sizes were used to study the interstitials content of the prepared TiAl alloys, especially oxygen content. Firstly, the $TiH_2$ and Al powders were blended in a mixer under Ar atmosphere for 180 min with a mass ratio of 66:34. Then, the mixed powder was respectively pretreated in a vacuum rotary furnace at 500, 600, 700 and 800 °C for 300 min with a heating rate of 5 °C min$^{-1}$. Subsequently, the Ti-48Al PA powder was crushed by a vibratory milling under vacuum until the mean powder grain size was about 15 μm. The particle size distributions were measured using the BT-9300S laser particle size distribution analyzer. These details would systematically discuss later.

**Table 1.** Particle size of the raw material powder.

| Powder | $TiH_2$ | Al | | |
|---|---|---|---|---|
| | | 1 | 2 | 3 |
| $D_{50}$ (μm) | 12 | 5.6 | 23.7 | 120.4 |

After that, the PA powder was compacted by cold isostatic pressing (CIP) at 200 MPa for 120 s. Then, the compacts were respectively sintered at 1300, 1350, 1400, and 1450 °C for 150 min with a heating rate of 3 °C·min$^{-1}$. Accordingly, these samples were referred to simply as T1300, T1350, T1400 and T1450. Finally, HIP was performed on the sintered T1350 and T1400 samples to obtain high relative density, and the two samples were marked

as H1350 and H1400. The HIP parameters were 1200 °C/140 MPa/120 min. The schematic diagram for the preparation process of PM TiAl alloy is presented in Figure 1.

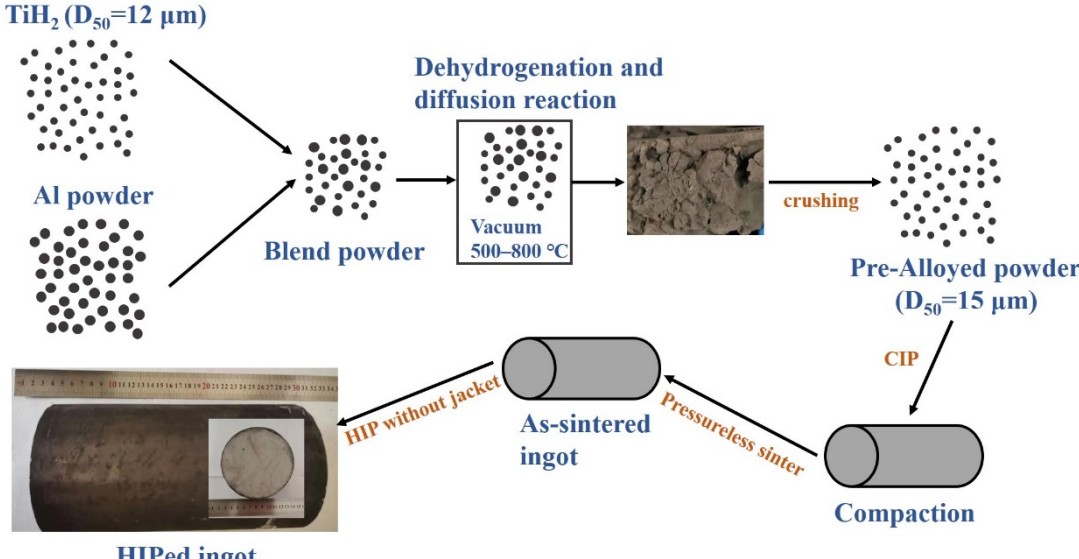

**Figure 1.** Schematic diagram for the preparation process of powder metallurgy (PM) TiAl alloy. HIP: hot isostatic pressing.

### 2.2. Testing and Characterization

Sintered densities were measured by the Archimedes method according to the ASTM standard B328. Phase constituents were analyzed by X-ray diffraction (XRD, D/max-RB, Japan) with Cu Kα radiation ($\lambda$ $\frac{1}{4}$ 0.15418 nm) at 50 kV and 200 mA with a diffraction angle 2θ from 10° to 90° and scanning rate of 5°·min$^{-1}$. The oxygen content of the samples was measured by non-dispersive infrared analysis, while nitrogen and hydrogen contents were determined by the thermal conductivity method, using an Eltra ONH-2000 apparatus. Powder morphology and as-sintered microstructures were analyzed by a scanning electron microscopy (SEM, Philips LEO-1450) equipped with energy-dispersive spectroscopy (EDS). Hardness was tested with a Vickers hardness tester (EM1500) at room temperature, and the loading pressure was 1 kg. Compression tests were carried out at room temperature using an INS-TRON 4206 machine. The test pieces were cylinders of 3 mm in diameter and 6 mm in length. The nominal strain rate for the compression tests was $10^{-3}$ s$^{-1}$. Room temperature tensile tests were conducted using an AGI-250 KN testing machine with a strain rate of $10^{-3}$ s$^{-1}$. Besides, element distribution was characterized by Electron Microprobe (EPMA, 1720).

## 3. Results and Discussion

### 3.1. PA Powder Analysis

In this study, the blended TiH$_2$ and Al powder was heated to a high temperature to form porous products, which took advantage of the Kirkendall effect between Ti and Al [11]. Then, the partially sintered porous products were crushed and milled to produce the non-spherical PA powder (~15 μm) with large specific surface area. At a high temperature, the possible reactions between TiH$_2$ and Al are shown in Equations (1)–(4) [32]:

$$6TiH_2 \rightarrow 6TiH_{1.5} + 1.5H_2 \rightarrow 6Ti + 6H_2 \tag{1}$$

$$6Ti + 6Al \rightarrow 4Ti + 2TiAl_3 \tag{2}$$

$$4Ti + 2TiAl_3 \rightarrow Ti_3Al + TiAl + 2TiAl_2 \tag{3}$$

$$2Ti_3Al + TiAl + 2TiAl_2 \rightarrow 6TiAl. \tag{4}$$

In order to select the appropriate reaction temperature, the Gibbs free energy ΔG needs to be calculated according to Equation (5):

$$\Delta G = \Delta H - T\Delta S. \tag{5}$$

In the formula, ΔH is the enthalpy change, T is the temperature, and ΔS is the entropy change.

Figure 2 shows the ΔG of Equations (1) and (2) with a function of temperature. The ΔG value was less than 0 when the temperature was above 500 °C, indicating that the reaction can proceed spontaneously since the ambient temperature was above 500 °C. Therefore, the minimum reaction temperature was set as 500 °C. The reaction of $TiH_2$ and Al included two reaction processes: Firstly, the endothermic reaction of $TiH_2$ decomposing into Ti and $H_2$. Secondly, the exothermic reaction of Ti and Al forming $TiAl_3$. When temperature was lower than 800 °C, the ΔG > 0 of $TiH_2$, the decomposition reaction indicated that the heat released by the formation of $TiAl_3$ between Ti and Al was absorbed by the decomposition of $TiH_2$.

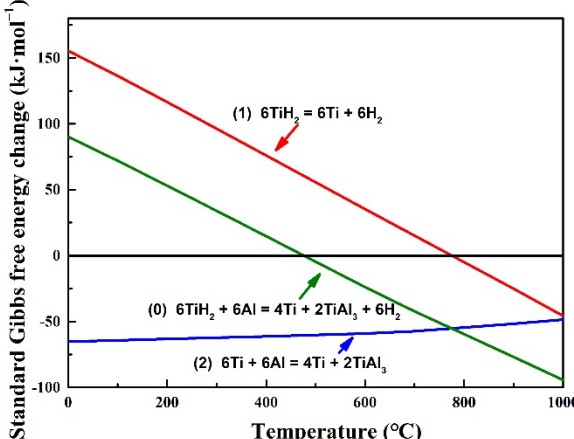

**Figure 2.** Gibbs free energy change (ΔG) with temperature of reactions (1) and (2).

It has been reported that Ti and Al powders synthesize TiAl by self-propagation high-temperature synthesis under pressure-less and high temperature conditions [33]. The reaction product has a non-metallurgical combination of layered structure with segregation. This is because the combustion wave takes place in the form of periodic oscillations or spins during the combustion process. In such cases, it is difficult to obtain micron-sized powder by mechanically crushing the product with a layered structure, which cannot be applied to the pressure-less sintering process. In order to utilize $TiH_2$ decomposition to absorb the heat released by the reaction of Ti and Al and prevent the self-propagating reaction, the upper limit of the reaction temperature was selected to be 800 °C.

The reactions in Equations (3) and (4) describe the diffusion process between $TiAl_3$ and Ti, during which $Ti_3Al$, TiAl, $TiAl_2$ were formed at the interface. When $TiAl_3$ was still present, TiAl and $TiAl_2$ diffuse competitively and grow up at the same time. When $TiAl_3$ was consumed, $Ti_3Al$ and $TiAl_2$ continued to decrease, and TiAl phase continued to increase. There were $Ti_3Al$ and TiAl phases in the final reaction product [34].

Based on the thermodynamic calculation results, the blended $TiH_2$ and Al powders were designed to pretreat at 500, 600, 700, and 800 °C. After treatment, the phases of the powders are shown in Figure 3a. At 500 °C, $TiH_2$ decomposed partially, and the main phases were $TiH_{1.5}$ and Al. There was no obvious diffraction peak of intermetallic compound. At 600 °C, Al had been completely consumed, and the main phases were γ-TiAl, $Ti_2Al$ and a small amount of α2-$Ti_3Al$ phase. At 700 °C, the $Ti_2Al$ phase content decreased, while the γ-TiAl and α2-$Ti_3Al$ phase content increased. Upon increasing to 800 °C, the reaction progressed completely, and only γ-TiAl and α2-$Ti_3Al$ phases existed.

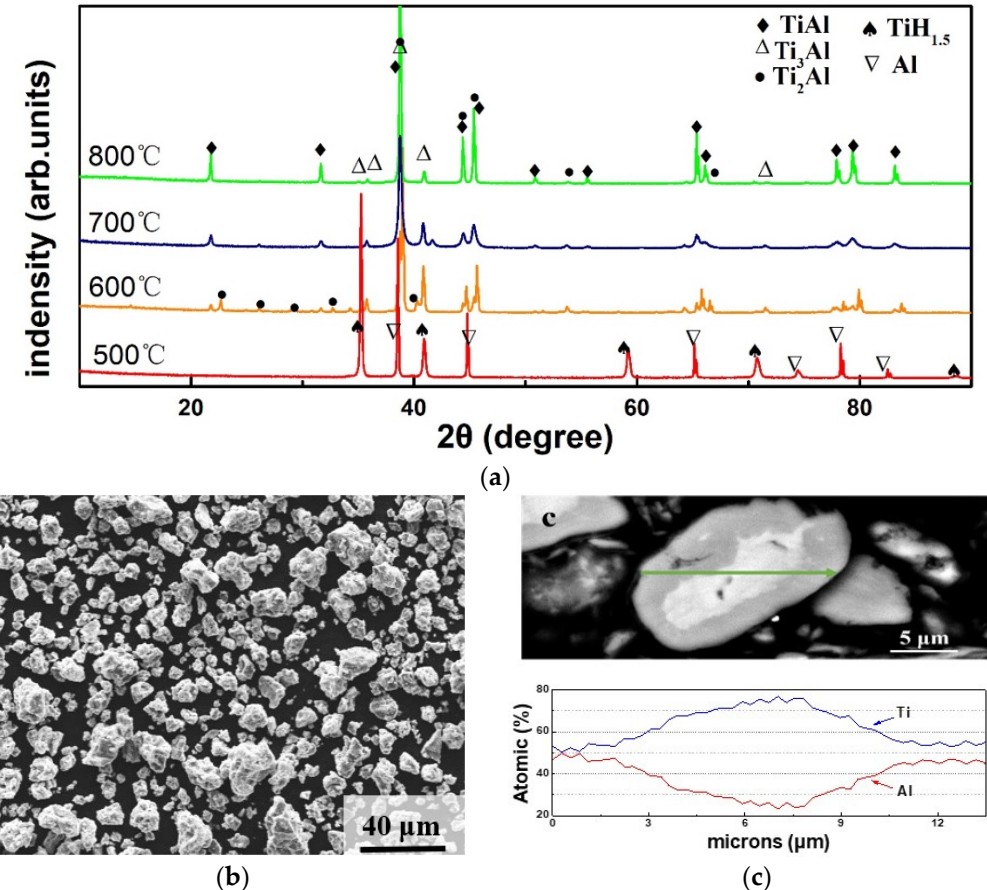

**Figure 3.** (**a**) X-ray diffraction (XRD) spectra of pre-alloyed powders prepared at different temperatures, (**b**) Pre-alloyed (PA) powder prepared at 800 °C, and (**c**) the cross section and composition distribution of the PA powder prepared at 800 °C.

In addition, Al powder was involved in the reaction as Equation (2). The process of forming $TiAl_3$ was an exothermic reaction with great volume expansion because of the Kirkendall effect, and it was difficult to achieve densification without pressure [19]. It can be inferred that the PA powder without Al was necessary for pressure-less sintering, and thus, the reaction temperature of 800 °C was suitable for PA powder preparation.

SEM image of the pretreated PA powder at 800 °C followed by crushing is shown in Figure 3b. It was found that the prepared PA powder was irregular. Figure 3c shows the cross-section and composition distribution of the PA powder. The PA powder was of "core-shell" structure, as shown in Figure 3c. According to the EDS results, the atomic ratio of aluminum and titanium in the outer layer was close to 1:1, while the core was enriched with Ti and leant with Al. Based on XRD analysis, the outer layer was γ-TiAl phase, while the core was $Ti_3Al$ phase. Moreover, multiple intermetallic layers might be formed in the PA powder, and diffusion between layers of various intermetallic compounds continued in the sintering process [35]. It was beneficial to achieve pressure-less sintering of the prepared PA powder since the diffusion between the intermetallic compound layers had little effect on the volume change. Besides, the homogenization of this alloy would be completed in the sintering process.

According to the Ti-Al phase diagram (shown in Figure 4) [4], the Ti-48Al alloy starts to solidify at 1456 °C. With cooling (top→bottom analysis), the composition of Ti-48Al alloy gradually narrows. The mechanism is the thermodynamic feasibility of the melting and solidification route [36,37]. The phase diagram confirms the stability of γ-TiAl at room temperature. Although the powder metallurgy route for producing Ti-Al alloy is not

shown in the equilibrium phase diagram, the mixed powder of titanium and aluminum alloy inherits the bottom→top direction.

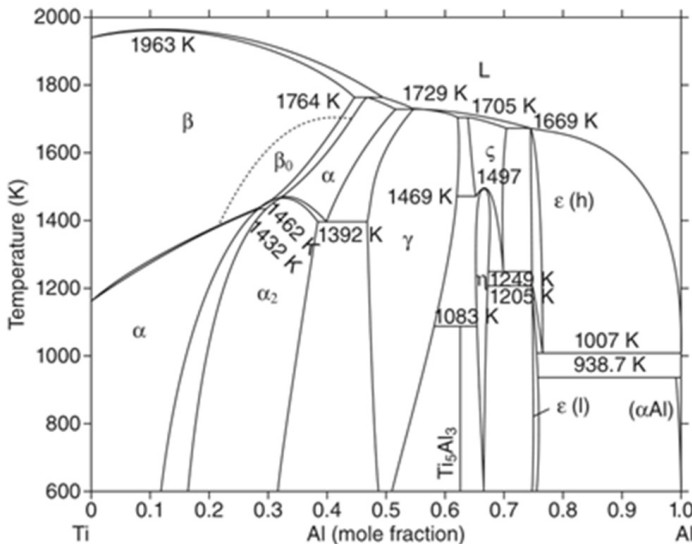

**Figure 4.** Al–Ti phase diagram (α: α-Ti, β: β-Ti, $\alpha_2$: $Ti_3Al$, γ: TiAl, ε: $TiAl_3$, L: liquid).

*3.2. γ-TiAl Alloy Preparation*

In addition to strict process control, the oxygen content was controlled by adjusting the particle size of Al powder. Table 2 shows the O, N, H contents of sintered samples at 1350 °C prepared with different Al powder particle sizes. As the particle size of the Al powder increased, the O content decreased. The sample prepared using coarse Al powder of 120 μm as raw material had the lowest oxygen content, about 1710 ppm. The specific surface area of coarse Al powder decreased, resulting in the absorbed oxygen content decreasing. There was an aluminum oxide film on the surface of Al powder, and the oxygen atom would dissolve into the matrix during the powder preparation and sintering process, which was difficult to eliminate.

**Table 2.** Chemical composition of sintered TiAl samples (wt.%, Bal. is short for balance).

| Particle Size of Al Powder (μm) | Ti (wt%) | Al (wt%) | O (wt%) | N (wt%) | H (wt%) |
|---|---|---|---|---|---|
| $D_{50}$ = 5.6 | Bal. | 33.9 | 0.505 | 0.028 | 0.005 |
| $D_{50}$ = 23 | Bal. | 34.5 | 0.324 | 0.022 | 0.006 |
| $D_{50}$ = 120.4 | Bal. | 34.1 | 0.171 | 0.019 | 0.005 |

In order to prepare γ-TiAl alloy with low oxygen content and high relative density, $TiH_2$ powder and coarse Al powder (~120 μm) were selected as final raw materials. Besides, all experimental operations were carried out under argon atmosphere. The PA powder was prepared at 800 °C and then compacted using cold isostatic pressing. Green body was vacuum pressure-less sintered at different temperatures. Finally, the residual pores were eliminated by HIP without steel capsule.

Sintering temperature was one of the important factors that affect the density. Relative densities of Ti-48Al alloy sintered at different sintering temperatures (1300–1450 °C) are shown in Figure 5. With the sintering temperature increasing, the relative density showed an upward trend, and the increasing amplitude gradually decreased. Compared to the T1400 sample, the relative density of T1450 had no obvious change. Therefore, when the sintering temperature was higher than 1400 °C, the temperature had less influence on the relative density.

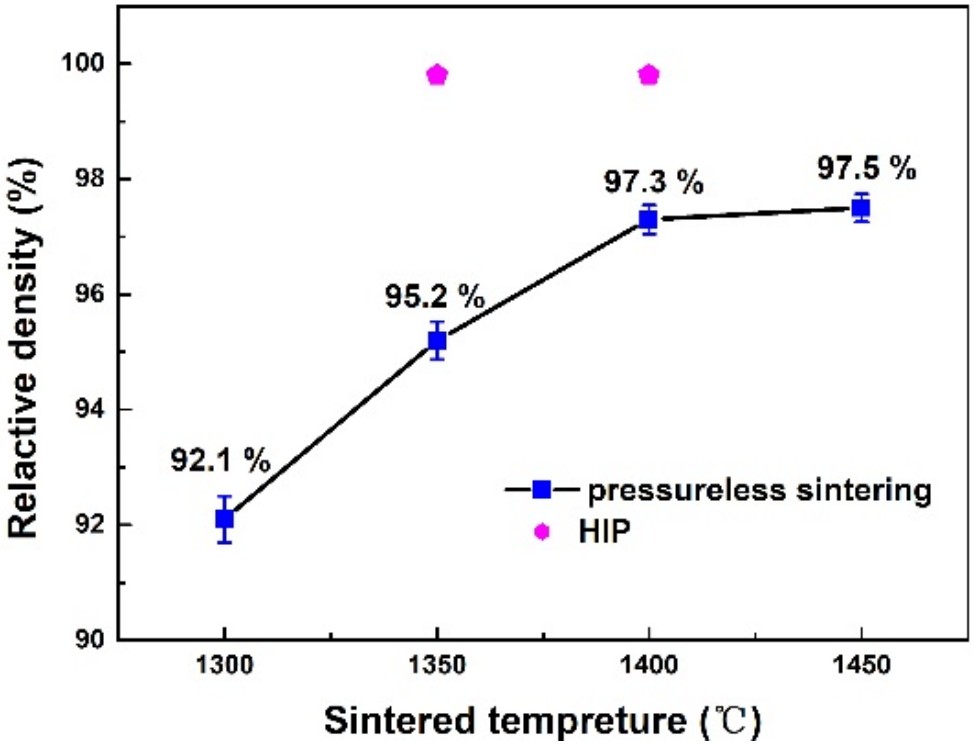

**Figure 5.** Relative densities of samples sintered at different temperatures.

In general, the sample prepared by HIP is developed to pack powder or green body into a capsule welded by steel or titanium plates [11,17]. Subsequent machining processing is required to remove the outer capsule, which is complicated with long process and high cost. For this reason, HIP without a capsule was employed to reduce the residual porosity. The pores in the sintered samples included open pores and closed pores. When the porosity was less than 5%, the proportion of open pores would be greatly reduced. Therefore, the sintered T1350 and T1400 samples with relative density of above 95% were selected for non-capsule HIP. The densities of the HIP processed samples are shown in Figure 6 marked as the "five-pointed stars". The relative density of H1350 and H1400 reached to 99.8%.

**Table 3.** Chemical composition of different regions marked in Figure 6 obtained by EDS analysis.

| Sample | Point | Al (at.%) | Ti (at.%) |
|--------|-------|-----------|-----------|
| T1300  | 1     | 37.85     | 62.15     |
|        | 2     | 48.57     | 51.43     |
| T1350  | 3     | 37.09     | 62.91     |
|        | 4     | 48.54     | 51.46     |
| T1400  | 5     | 37.61     | 62.39     |
|        | 6     | 36.87     | 63.13     |
|        | 7     | 48.60     | 51.40     |
| T1450  | 8     | 39.74     | 60.26     |
|        | 9     | 39.50     | 60.50     |
|        | 10    | 48.79     | 51.21     |

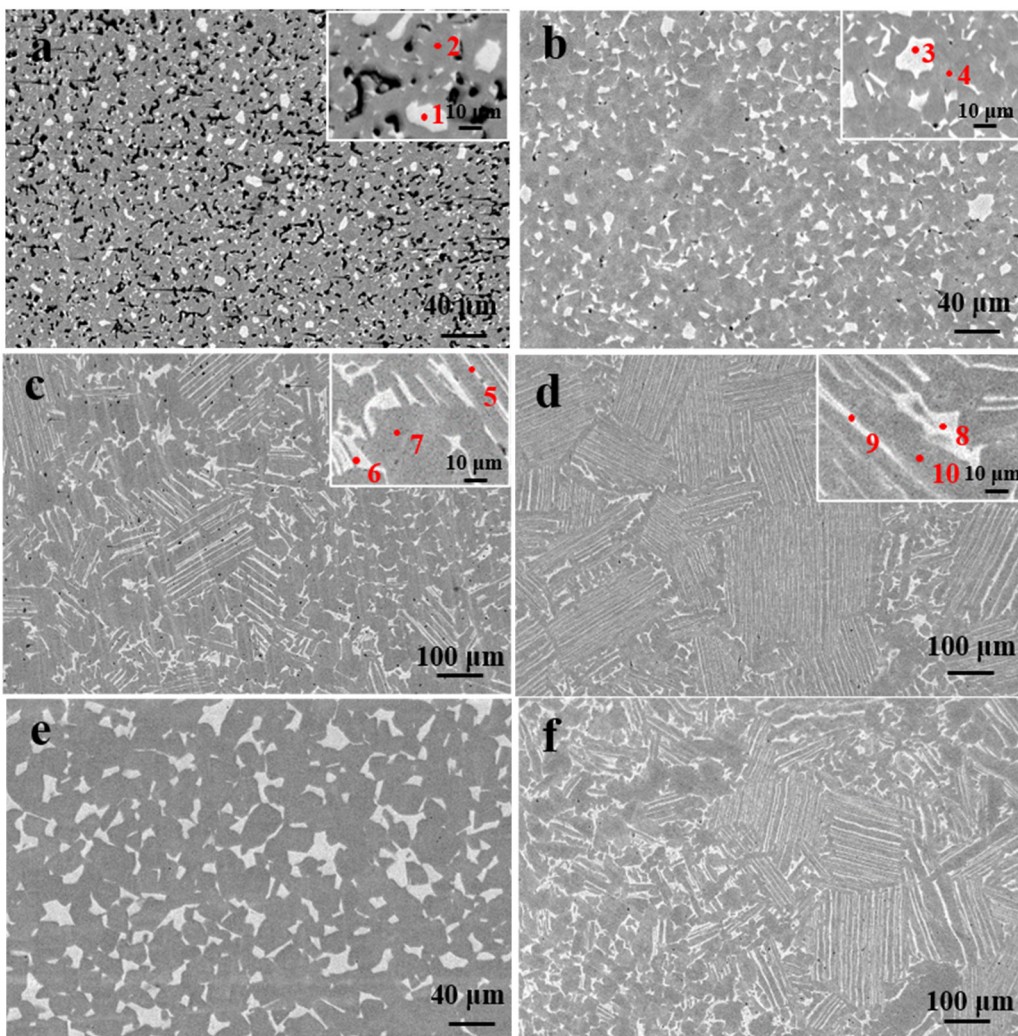

**Figure 6.** SEM images of as-sintered samples: (**a**) 1300 °C, (**b**) 1350 °C, (**c**) 1400 °C, (**d**) 1450 °C (**e**) 1350 °C/HIP, and (**f**) 1400 °C/HIP. The marked red numbers are corresponding to the measured EDS positions shown in Table 3.

### 3.3. Phase and Microstructure

The microstructures of the samples, sintered at different conditions, are shown in Figure 6. There were a large number of pores on the prior particle boundaries in the sample sintered at 1300 °C, which was in good agreement with the relative density result. As the sintering temperature increased, the pores gradually decreased. At 1450 °C, only a few pores remained in the sample.

Combined the EDS results shown in Table 3 with the XRD results, it can be known that the gray area in Figure 6 was the $\gamma$-TiAl phase and the bright area was the $\alpha2$-Ti$_3$Al phase. The microstructure of the T1350 sample was near-$\gamma$. Near-$\gamma$ was composed of equiaxed $\gamma$ particles and a small number of $\alpha2$ particles distributed in the $\gamma$ grain boundaries [35–37]. The microstructure of T1400 sample was duplex, as shown in Figure 6c. The duplex structure was composed of equiaxed $\gamma$ grains and $\gamma/\alpha2$ layers, in which the $\alpha2$ phase was precipitated from the disordered $\alpha$ phase existing at high temperature. The grain size of duplex was generally small because the $\alpha$ phase and $\gamma$ phase were pinned to each other at high temperatures, resulting in slow grain growth. Moreover, the dispersed $\alpha2$ phase in duplex would increase the strength with the sacrifice of plasticity [2,38]. The microstructure of the T1450 sample was nearly lamellar, as shown in Figure 6d. Nearly lamellar was composed of $\gamma/\alpha2$ layers and a small number of equiaxed $\gamma$ grains distributed between the layer clusters. As the $\gamma$ phase was reduced, the pinning effect on the $\alpha$ phase

was weakened, resulting in larger lamellae. This kind of microstructures had high fracture toughness and creep resistance, but also showed low ductility at room temperature [2].

After HIP, the corresponding microstructures with high relatively density were observed. Comparing Figure 6b,c,e,f, it was found that HIP did not change the type of microstructure, and the pores reduced significantly. Although dense microstructure was observed in the HIP samples, grain growth was also inevitable under this HIP condition. Compared to Figure 6b,c, coarse microstructure was obviously found in Figure 6e,f.

Accordingly, the element distribution of H1350 sample was analyzed (Figure 7). In the γ phase and α2 phase, the distribution of Ti and Al was relatively uniform. It could be proven that the problem of multilayer intermetallic compounds in the prepared PA powder (as seen in Figure 3c) had been solved.

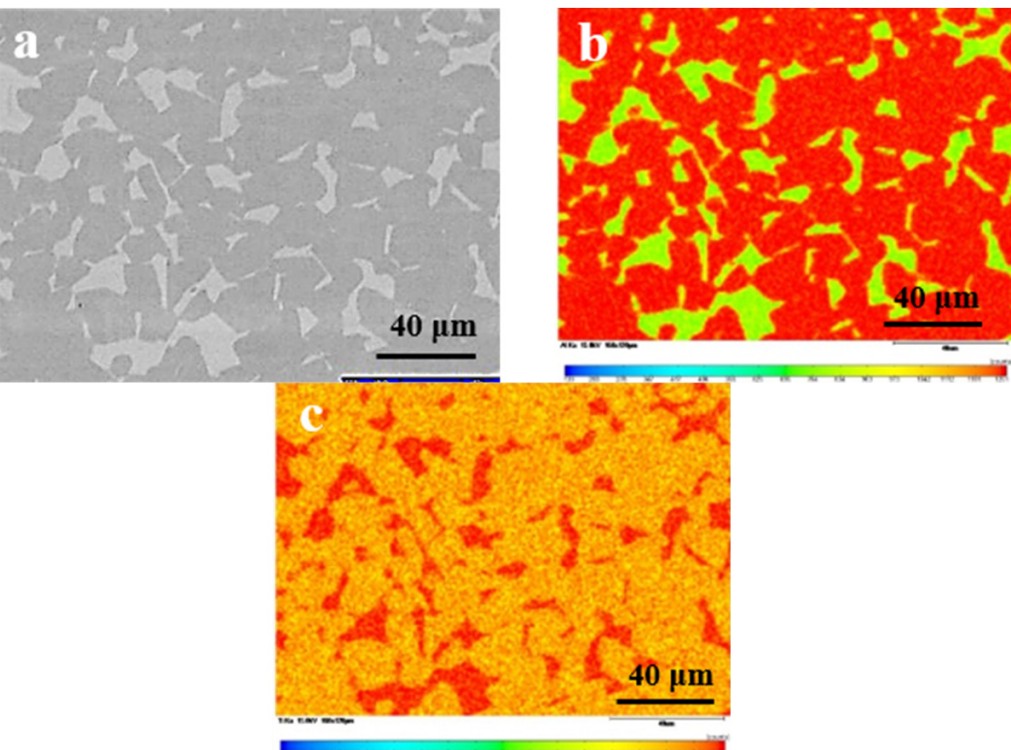

**Figure 7.** (**a**) SEM image, (**b**) Al, and (**c**) Ti element distribution maps of H1350 sample.

### 3.4. Mechanical Properties

The relationship between Vickers hardness and sintering temperature is shown in Figure 8. The hardness value of as-sintered compact increased with the increasing of sintering temperature, but decreased after 1400 °C. The reason for the hardness increasing was that the relative density increased and the porosity decreased with the temperature increasing at 1300–1400 °C. Upon a further increase in the temperature to 1450 °C, the hardness value decreased due to the coarse grains and the transformed structure from duplex to sub-lamellar structure. High sintering temperature promoted unnecessary grain growth, which was harmful to mechanical properties. Therefore, 1350 °C and 1400 °C were selected as appropriate sintering temperatures for PM Ti-48Al alloys. Compared with as-sintered samples, the HIP samples had lower hardness, as shown in Figure 8a. Under the action of high temperature (1200 °C) and high pressure (140 MPa), the grains significantly grew and the structure became coarser.

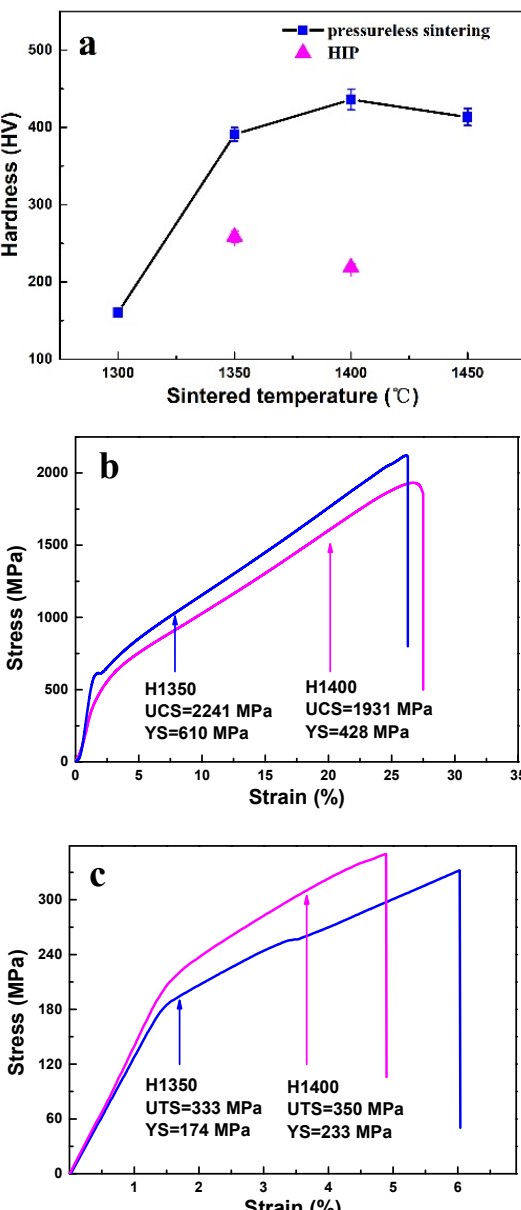

**Figure 8.** (**a**) Hardness, (**b**) compression properties, and (**c**) tensile properties of the prepared PM Ti-Al alloys. UCS: ultimate compressive strength, YS: yield strength, UTS: ultimate tensile strength.

Figure 7b presents the compression properties of these two HIP samples. Besides, Table 4 compares the compressive strengths of the HIP Ti-48Al alloys and some alloys reported in related literatures [39–41]. The compressive strength of H1300 sample was 2241 MPa, while that of the H1400 sample was 1931 MPa. Compared with the as-cast Ti-48Al-2Nb-2Cr alloy [40], the compressive strength of the prepared Ti-48Al alloy increased by 130% although there were no two alloying elements (Nb and Cr), which proved that the prepared PM Ti-Al alloys, using the new kind of PA powder, had higher properties. Furthermore, TiAl alloys prepared by traditional casting method would form coarse columnar crystals during the cooling process, because of the poor thermal conductivity of TiAl alloys. With the powder metallurgy process of γ-TiAl alloy, a very fine and chemically uniform microstructure could be obtained, which was beneficial for mechanical properties. In addition, compared with the Ti-50Al alloy prepared by hot pressing [39], the prepared Ti-48Al alloy had a 72% increase in the compressive strength.

**Table 4.** Compression properties of TiAl alloys in this work and other reports.

| Alloy (at%) | Processing | Compressive strength (MPa) | Reference |
|---|---|---|---|
| Ti-48Al | Sintering + HIP(H1350) | 2241 ± 52 | This work |
| Ti-48Al | Sintering + HIP(H1400) | 1931 ± 17 | This work |
| Ti-50Al | Hot pressing | 1300 | [34] |
| Ti-50Al-10wt%(Nb-C) | Hot pressing | 1649 | [34] |
| Ti-48Al-2Nb-2Cr | Cast | 950 | [35] |
| Ti-48Al-2Nb-2Cr-2Cu | sintering | 1700 | [36] |

The HIP samples were subjected to a tensile test at room temperature. The tensile strength of the H1350 sample was 333 MPa, while that of the H 1400 sample was 350 MPa. Correspondingly, the tensile fracture morphology was observed, as shown in Figure 9. These two alloys had good densification state with fewer micro-cracks. The fracture was a brittle fracture. The main fracture modes in Figure 9a were consisted with a transcrystalline fracture and intergranular fracture. Besides, the fracture modes in Figure 8b included transcrystalline, intercrystalline, translamellar and interlamellar fractures. The fracture modes of these two alloys were different because the structures were different, as shown in Figure 6e,f. Compared with the H1350 sample with near-$\gamma$ microstructure, the H1400 sample had duplex microstructure with near-$\gamma$ and lamellar structure, and thus the fracture appeared as translayer fracture and interlayer fracture. In addition, the cleavage facets were smaller in Figure 9a. This was because the grains became coarser with the sintering temperature increasing.

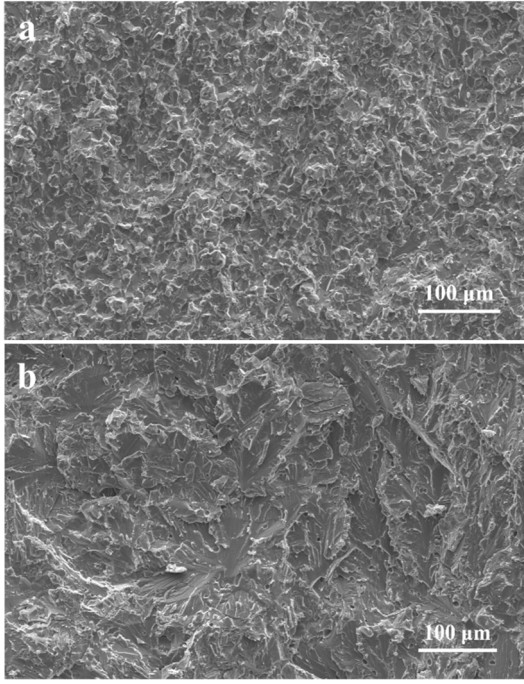

**Figure 9.** Fracture morphology of the prepared TiAl alloys (**a**) H1350, (**b**) H1400.

### 4. Conclusions

A new kind of non-spherical PA TiAl powder was prepared by the reaction of TiH$_2$ with Al at high temperature, and then Ti-48Al alloy with high relative density and low oxygen content was prepared by cold isostatic pressing and vacuum pressure-less sintering. These conclusions were as follows:

(1) Using the blended TiH$_2$ and Al powders as raw materials, a non-spherical TiAl pre-alloy powders were prepared by reacting at high temperature followed by crushing. There were only $\gamma$-TiAl and $\alpha$2-Ti$_3$Al phases in the PA powder pretreated at 800 °C.

(2) The irregular pre-alloyed powder had good sinterability. After pressure-less sintering at 1350 °C and 1400 °C temperatures, the relative densities were 95.2% and 97.3%, respectively. Followed by non-capsule HIP, both relative densities increased to nearly 100%.

(3) The particle size of the raw Al powder significantly affected the oxygen content of the sample. The larger the Al particle size, the smaller the O content. The sample prepared with coarse Al powder of 120 μm had a low oxygen content of 1710 ppm.

(4) The as-sintered samples had a near-γ structure at 1350 °C a duplex structure at 1400 °C, and a nearly lamellar structure at 1450 °C. Besides, the sample sintered at 1350 °C followed by HIP had comprehensive properties with compressive strength of 2241 MPa and tensile strength of 333 MPa.

**Author Contributions:** Conceptualization, Z.G.; methodology, Z.G.; validation, M.Y. and F.Y. and Z.G.; investigation, M.Y., F.Y., B.L. and C.C.; resources, Z.G. and Y.S.; writing—original draft preparation, M.Y.; writing—review and editing, M.Y. and F.Y.; project administration, C.C. and F.Y.; funding acquisition, Z.G., C.C. and F.Y. All authors have read and agreed to the published version of the manuscript.

**Funding:** This study was funded by the National Natural Science Foundation of China (No. 52004027), the Fundamental Research Funds for the Central Universities (No. FRF-GF-20-05A), the Innovation Group Project of Southern Marine Science and Engineering Guangdong Laboratory (Zhuhai) (No. 311020012), and the State Key Lab of Advanced Metals and Materials (No. 2020-Z17).

**Institutional Review Board Statement:** The study did not involve humans or animals.

**Informed Consent Statement:** The study is not involving humans.

**Data Availability Statement:** The data used in this article are presented in the manuscript.

**Conflicts of Interest:** The authors declare no conflict of interest. The funders had no role in the design of the study; in the collection, analyses, or interpretation of data; in the writing of the manuscript, or in the decision to publish the results.

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
