# Peer review of "Microstructure and Mechanical Properties of High Relative Density γ-TiAl Alloy Using Irregular Pre-Alloyed Powder"

_metals, doi:10.3390/met11040635_

Round 1

Reviewer 1 Report

The authors do not fully account for the H2 in the system, since there is no characterization, an explanation is required. The XRD peaks of  TiH2 were detected at 500 C, how was H2 measured?

The authors must fully explain the mechanism of alloy formation. For reference see the following article"Investigation into the thermal behaviour of the B2–NiAl intermetallic alloy produced by compaction and sintering of the elemental Ni and Al powders,Vacuum 169, 108919. 

Author Response

Reviewer #1:

1)The authors do not fully account for the H2 in the system, since there is no characterization, an explanation is required. The XRD peaks of TiH2 were detected at 500 C, how was H2 measured?

Authors’ response:

It is very kind of you to give us direction. TiH2 is used as a raw material, and the hydrogen content of TiH2 is about 4wt.%. Through thermodynamic calculations, it is found that when the temperature is ≥500℃, the Gibbs free energy of the decomposition reaction of titanium hydride is less than 0, indicating that this is a spontaneous reaction. And the TiH1.5 appears in the XRD result at 500 °C, which is a good indication that the titanium hydride has begun to decompose. The hydrogen content of the sintered compact is tested. In Table 2, the hydrogen content is less than 0.006wt.%, which indicates that a large amount of hydrogen escapes in the form of hydrogen during the powdering and sintering process. In addition, in our previous work (Reference 28), Ti-6Al-4V was prepared using the HDH method, which uses the characteristic that TiH2 can decompose into Ti and hydrogen at high temperatures.

2)The authors must fully explain the mechanism of alloy formation. For reference see the following article"Investigation into the thermal behaviour of the B2–NiAl intermetallic alloy produced by compaction and sintering of the elemental Ni and Al powders,Vacuum 169, 108919.

Authors’ response:

Considering the reviewer’s suggestion, we refer to the above-mentioned documents and add some descriptions. The added content part is in line 177 on page 6, as follows:

According to the Ti-Al phase diagram in Fig. 4, the Ti-48Al alloy starts to solidify at 1456°C. With cooling (top→bottom analysis), the composition of Ti-48Al alloy gradually narrows. The mechanism is the thermodynamic feasibility of the melting and solidification route [35]. The phase diagram confirms the stability of γ-TiAl at room temperature. Although the powder metallurgy route for producing Ti-Al alloy is not shown in the equilibrium phase diagram, the mixed powder of titanium and aluminum alloy inherits the bottom→top direction.

Reviewer 2 Report

See attached file. you can make a better explanation of superalloys and Ti alloys, in addition to attached file

Author Response

1) You must acknowledge it with a sound literature review, looking every place where the ideas are presented, now you only search in alloys. For aerospace and automobile industries [1-4], and Beranoagirre ideas missed, pity.

Authors’ response:

Thank you very much for the reviewer’s suggestions. We have rewritten the introduction and referred to Beranoagirre ideas. The rewritten part is in line 28 on page 1, as follows:

In the aerospace industry, γ-TiAl can be used instead of Inconel 718 to prepare components such as turbine blades and compressor blades. In the automotive industry, these materials are suitable for racing and high-end vehicle parts such as engine valves, turbine wheels and connecting rods [5-6]. For example, Ti-48Al-2Cr-2Nb alloy has been used to manufacture the six-stage and seven-stage low pressure turbine blades in GEnx engines [7]. However, TiAl alloys generally suffer from low ductility at room temperature, resulting in the manufacturing difficulties. Some reports have found that adjusting the grain size in a suitable range and improving the homogeneity are beneficial to improve the workability of this alloy [8-10].

2) HIP…how much is the size able for components.

Authors’ response:

The hot isostatic pressing equipment we used can accommodate samples with a height of 230 mm and a diameter of 150 mm. The size is enough to prepare some parts. Regarding large-size parts, large-size HIP equipment can be used. At present, large-size HIP equipment has been widely employed to produce different materials. In this study, non-capsule HIP was employed, which significantly decreases the production cost and simply the production process. Besides, we are also trying to prepare high relative density alloys without HIP by the following methods: 1) Reduce the size of the pre-alloyed powder, so that the powder has a larger specific surface area and sintering driving force; 2) The introduction of some alloying elements can improve the performance while also promote the sintered density.

3) Do you know that some engine manufacturers have stop the use of gamma TiAl for aero engines due to fragility?

If you think in the life of a blade…it is a hell, centrifugal foces are too high.

Authors’ response:

Thanks very much for the suggestions. The plasticity and fragility are indeed important issues affecting the application of this alloy. A lot of researches are also devoted to finding ways to improve plasticity. One of the development trends of TiAl-based alloy is to obtain a fine and uniform alloy structure to balance and improve its comprehensive mechanical properties. Some studies have shown that the plasticity and strength can be improved by refining the grains. It can be seen from the figure below that as the grain size increases, the elongation decreases rapidly.

There are mainly the following grain refinement technologies: third element alloying, solid phase transformation, phase solidification refinement, thermomechanical treatment and non-equilibrium processing technology, etc. Some of our following work is also working towards improving the plasticity of this alloy.

Round 2

Reviewer 2 Report

Ok